# Green Synthesis of Silver and Gold Nanoparticles via *Sargassum serratifolium* Extract for Catalytic Reduction of Organic Dyes

**Beomjin Kim** [1,†] , **Woo Chang Song** [1,†] , **Sun Young Park** [2,*] and **Geuntae Park** [1,*]

1    Department of Nanofusion Technology, Pusan National University, Busan 46241, Korea;
     201210503@pusan.ac.kr (B.K.); dck3202@naver.com (W.C.S.)
2    Bio-IT Fusion Technology Research Institute, Pusan National University, Busan 46241, Korea
*    Correspondence: sundeng99@pusan.ac.kr (S.Y.P.); gtpark@pusan.ac.kr (G.P.); Tel.: +82-51-510-3630 (S.Y.P.);
     +82-51-510-3740 (G.P.); Fax: +82-51-514-7065 (S.Y.P.); +82-51-518-4113 (G.P.)
†    These authors contributed equally to this work.

**Abstract:** The green synthesis of inorganic nanoparticles (NPs) using bio-materials has attained enormous attention in recent years due to its simple, eco-friendly, low-cost and non-toxic nature. In this work, silver nanoparticles (AgNPs) and gold nanoparticles (AuNPs) were synthesized by the marine algae extract, *Sargassum serratifolium* (SS). The characteristic studies of bio-synthesized SS-AgNPs and SS-AuNPs were carried out by using ultraviolet–visible (UV–Vis) absorption spectroscopy, dynamic light scattering (DLS), high-resolution transmission electron microscope (HR-TEM), selected area electron diffraction (SAED), energy-dispersive X-ray spectroscopy (EDX), X-ray powder diffraction (XRD) and Fourier transform infrared spectroscopy (FT-IR). Phytochemicals in the algae extract, such as meroterpenoids, acted as a capping agent for the NPs' growth. The synthesized Ag and Au NPs were found to have important catalytic activity for the degradation of organic dyes, including methylene blue, rhodamine B and methyl orange. The reduction of dyes by SS-AgNPs and -AuNPs followed the pseudo-first order kinetics.

**Keywords:** catalytic activity; gold nanoparticles; green synthesis; *Sargassum serratifolium*; silver nanoparticles





## 1. Introduction

Water pollution caused by industrialization is one of the major problems of environmental pollution. Synthetic dye contained in wastewater is one of the factors that have many influences on water pollution. Dyes are widely used in various fields such as textiles, medical care, cosmetics, ink, plastics, and pharmaceuticals. In these industries, it is very important for the environment and people's health to decompose and cleanly treat dyes with toxic, mutagenic and carcinogenic properties [?].

Until now, a method of using nanoparticles as a catalyst for degrading dyes has been studied by many researchers. Metal nanoparticles (MNPs) have received attention in extensive modern research fields, including electronics, energy, catalysis, environment, sensors, optics, and medicine [?]. MNPs that have a high ratio of surface area have their own optical and chemical characters. Many different MNPs, such as silver (AgNPs), gold (AuNPs), platinum (PtNPs), palladium (PdNPs) and iron nanoparticles, have been synthesized and investigated in various ways including photochemical synthesis [?], electrochemical [?] and reduction of solution [?] etc. In particular, AuNPs have been applied for drug delivery, biomarkers in diagnosis, sensors and energy generation, while AgNPs have been used in textiles, electronic products, biosensors and biological industries with their antibacterial, anti-inflammatory and antitumor activities [? ? ? ?].

As the process evolves, some toxic chemicals are primarily used as reducing agents in the traditional process, causing environmental pollution. Chemicals employed for

formation of NPs pollute the environment by creating pollutants, which are harmful to the aquatic environment and human health. Also, the traditional methods for synthesis of MNPs have a complicated process consuming a lot of cost, energy and time.

To address these shortcomings, many studies have been reported on green synthesis with antibacterial, neuroprotective, anticancer and photocatalytic activity in recent years [**? ? ? ?** ]. Green synthesis aims to avoid producing products that are harmful to the environment through a reliable, sustainable and eco-friendly synthesis process [**?** ]. Therefore, in the green synthesis of metal nanoparticles, microorganisms (e.g., bacteria, algae, and fungi) and plant extracts (e.g., leaves, root, flower, and fruit) are used as reducing agents. The method of forming MNPs using biological materials is simple, inexpensive, eco-friendly, low energy and non-toxic with high stability.

*Sargassum serratifolium* (SS), a brown alga, is a member of genus *Sargassum* of the family *Sargassaceae*. It is widely distributed along the coasts of Korea and Japan and has been used as a food and traditional medicine in Korea and China for a long time [**?** ]. In recent years, various physiological activities (anti-oxidant, neuroprotective, anti-inflammatory, anti-cancer, hepatoprotective and hypopigmentation etc.) have been reported from in vivo and in vitro studies using SS extracts [**? ? ? ? ? ?** ]. It has been known that *Sargassum* species have bioactive molecules, like meroterpenoids, phlorotannins, fucosterols and fucoxanthins. SS, in particular, contains meroterpenoids in high concentration, which are sargachromenol (SCM), sargaquinoic acid (SQA), and sargahydroquinoic acid (SHQA). They are major active ingredients with antioxidant and anti-inflammatory properties [**? ? ?** ]. Therefore, we synthesized silver and gold nanoparticles applying these active ingredients from the SS extract in this study for reducing organic dyes. We designated these nanoparticles as SS-AgNPs and SS-AuNPs. These NPs were characterized employing ultraviolet–visible (UV–Vis) spectroscopy, energy-dispersive X-ray spectroscopy (EDX), X-ray diffraction (XRD) and Fourier transform infrared spectroscopy (FT-IR). Dynamic light scattering (DLS) and high resolution-transmission electron microscopy (HR-TEM) were employed to measure size distribution, zeta potential and shape. This is the first study of inorganic NPs green synthesized using SS extract and their evaluation in the catalytic reduction of various synthetic dyes, including methylene blue, rhodamine B and methyl orange, using $NaBH_4$.

## 2. Results and Discussion

### 2.1. Bio-Fabrication of Sargassum Serratifolium Nanoparticles (SS-NPs)

Green synthesized SS-AgNPs and SS-AuNPs were characterized to investigate their potential (Figure **??**). We mixed 1 mL aqueous extract of SS (2 mg/mL) with 1 μL $AgNO_3$ (1 M) solution. The color of extract turned to dark orange color within 15 min (Figure **??**b). This change of color appeared by localized surface plasmon resonance (LSPR). UV–Vis spectra at 300–800 nm also confirmed the formation of SS-NPs due to LSPR. A maximum LSPR band at 422 nm defined the bio-fabrication of SS-AgNPs. Similarly, 1 mL aqueous extract of SS (2 mg/mL) was added to 1 μL $HAuCl_4 \cdot 3H_2O$ (1 M) solution for SS-AuNPs' formation. The color of the colloid became violet within 15 min in the water bath, suggesting SS-AuNPs had formed (Figure **??**c). SS-AuNPs showed a strong LSPR band at 528 nm shown in the spectra. To the best of our knowledge, it was the first report on green-synthesis of noble NPs with SS, although many previous researchers have reported bio-formation of noble NPs [**?** ]. Ramkumar et al. [**?** ] studied that biofabrication of silver NPs using *Enteromorpha compressa* substantiated an LSPR peak at 421 nm and other red, green, and brown algae can also be used successfully as a reductant agent to synthesize inorganic nanoparticles.

### 2.2. Dynamic Light Scattering (DLS) and High-Resolution Transmission Electron Microscopy (HR-TEM) Analysis

Figure **??** shows the DLS analysis to find out the hydrodynamic distribution of size for SS-AgNPs and -AuNPs. The average size of SS-AgNPs that DLS measured was

120.0 ± 3.85 nm and that of SS-AuNPs was 17.47 ± 0.13 nm at 25 °C. The polydispersity index was 0.397 and 0.744, respectively. The zeta potential measurements that identify the stability of nanoparticles were −34.3 ± 0.47 and −31.9 ± 0.75 mV, respectively as shown in Figure **??**. These large negative values of zeta potential indicated a good stability of the colloidal solutions of SS-AgNP and -AuNP.

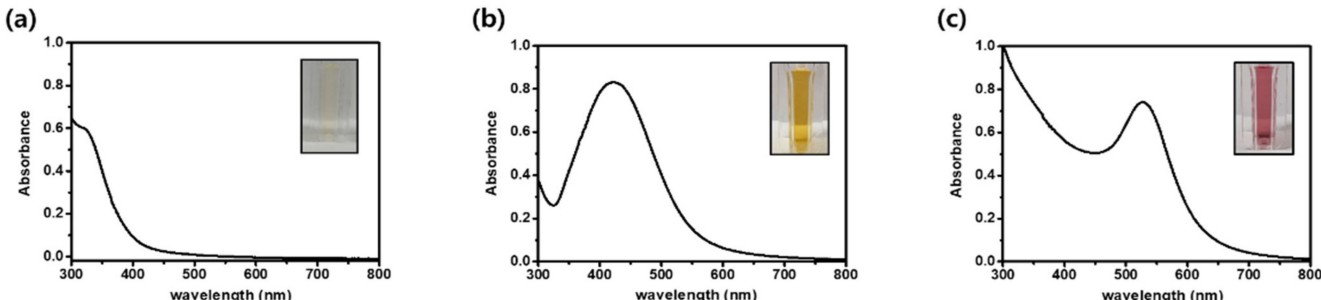

**Figure 1.** Ultraviolet–visible (UV–Vis) absorption spectra of (**a**) *Sargassum serratifolium* (SS) extract, (**b**) SS-AgNPs (SS-silver nanoparticles) and (**c**) SS-AuNPs (SS-gold nanoparticles).

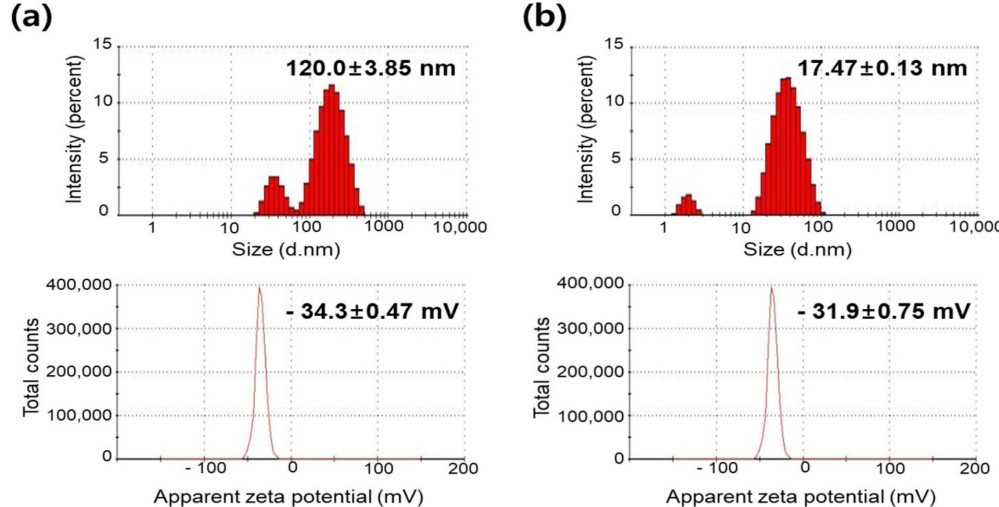

**Figure 2.** Dynamic light scattering (DLS) showing the colloidal size and Zeta potential of (**a**) SS-AgNPs and (**b**) SS-AuNPs.

HR-TEM was employed to determine the morphology, dispersion, and shape of the green synthesized NPs. **????** exhibit the TEM images for SS-AgNPs and SS-AuNPs. Most of the Ag and Au NPs were slightly spherical in shape but other shapes, like triangles, pentagons, and narrow square, were also observed. The average sizes of silver and gold nanoparticles shown in the TEM image are about 27.84 and 7.18 nm, respectively. The selected area electron diffraction (SAED) pattern in Figures **??**c and **??**c corroborated that the formed nanoparticles had crystalline structure in nature. The bright circles corresponded to (111), (200), (220) and (311) lattice planes each, indicating an FCC (face-centered cubic) structure of SS-NPs.

The EDX spectra of the NPs are shown in Figure **??**f, SS-AgNPs exhibited the highest optical absorption peak at 3 keV, and SS-AuNPs displayed peaks at 2–2.5 and 9.5–10 keV. These results were typical of those of previous reports of green-synthesized AgNPs and AuNPs [**? ?** ].

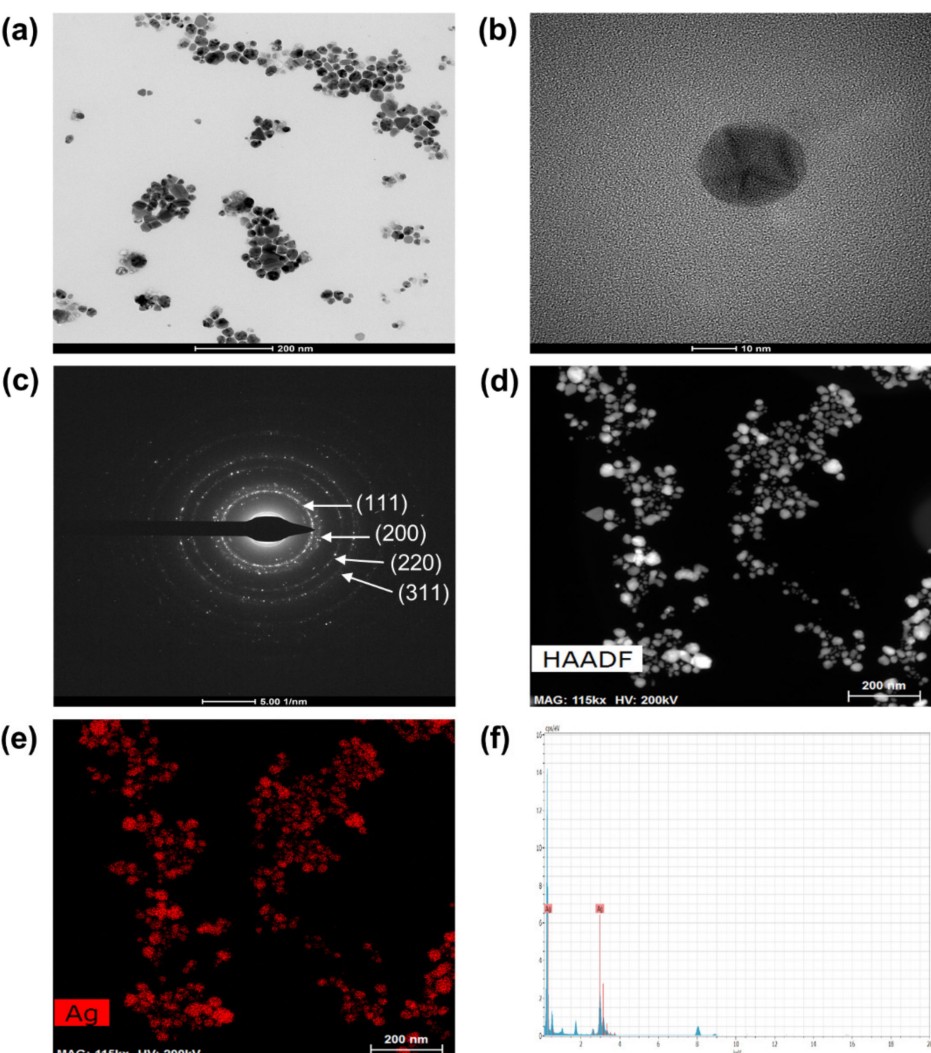

**Figure 3.** High-resolution transmission electron microscopy (HR-TEM) images (**a**,**b**), selected area electron diffraction (SAED) pattern (**c**), HAADF (High-angle annular dark-field) images (**d**,**e**) and energy-dispersive X-ray spectroscopy (EDX) spectra (**f**) of SS-AgNPs. 58 k (**a**), 630 k (**b**), 115 k (**d**,**e**).

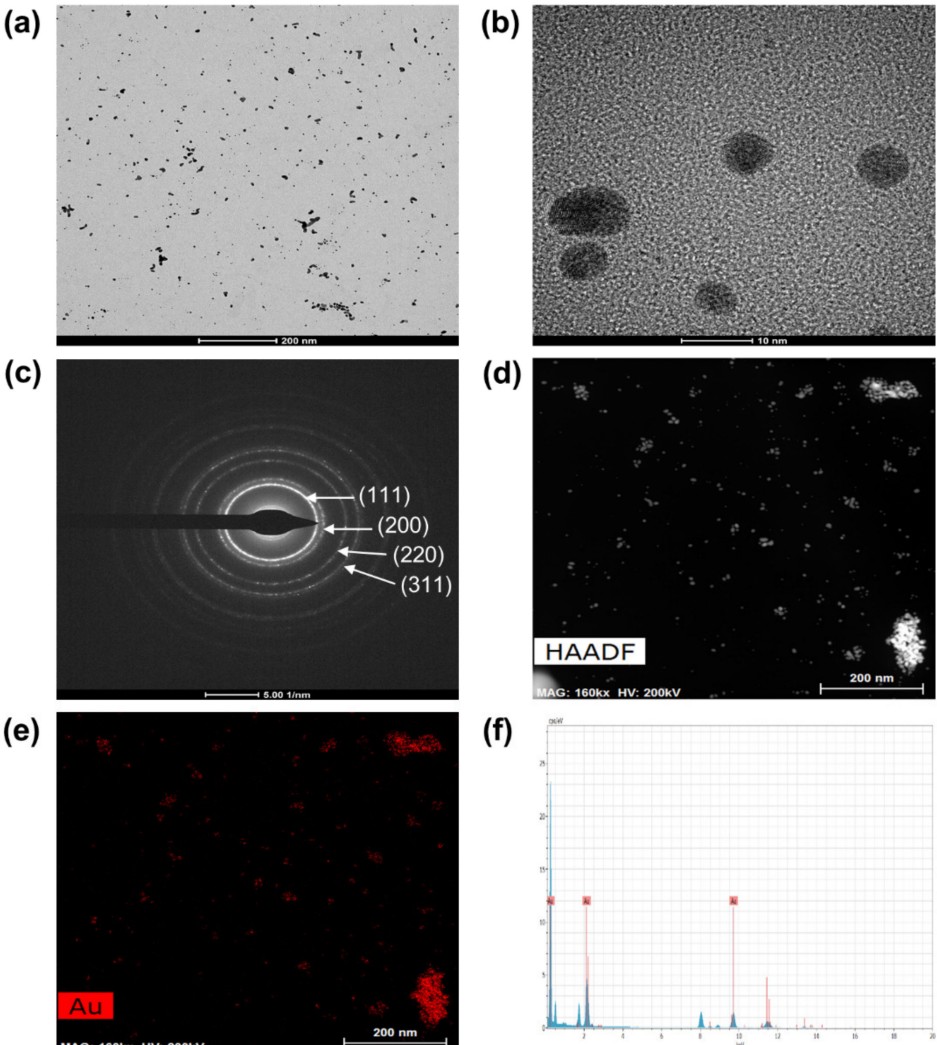

**Figure 4.** HR-TEM images (**a**,**b**), SAED pattern (**c**), HAADF images (**d**,**e**) and EDX spectra (**f**) of SS-AuNPs. 58 k (**a**), 630 k (**b**), 115 k (**d**,**e**).

### 2.3. X-ray Diffraction (XRD) Analysis

XRD analysis revealed the crystalline properties of bio-fabricated SS-AgNPs and SS-AuNPs. Figure **??** exhibits the XRD patterns of SS-AgNPs and -AuNPs. Bragg's reflections peaks of SS-AgNPs at 2θ values 38.0°, 44.1°, 64.4°, and 77.3°, which indexed to (111), (200), (220) and (311) planes each. In SS-AuNPs, the bands at 38.3°, 44.4°, 64.8°, and 77.7° also corresponded to the same planes in SS-AgNPs. Thus, the XRD analysis confirmed both SS-AgNPs and SS-AuNPs have face-centered cubic structures. The crystalline sizes of the AgNP and AuNP was obtained from the Debye–Scherrer formula,

$$D = \frac{k\lambda}{\beta \cos\theta} \qquad (1)$$

where $D$ is the average crystalline size of the NPs, $k$ is 0.9, Scherrer constant, $\lambda$ is the 0.15406, wavelength of the x-ray sources and $\beta$ is the full-width at half maximum (FWHM) of the peak in XRD spectra at Bragg angle, $\theta$. The average crystalline sizes of SS-AgNP and SS-AuNP were determined as ~9.39 and ~5.22 nm, respectively. The sizes were in agreement with those shown in the HR-TEM image (Figures **??**b and **??**b).

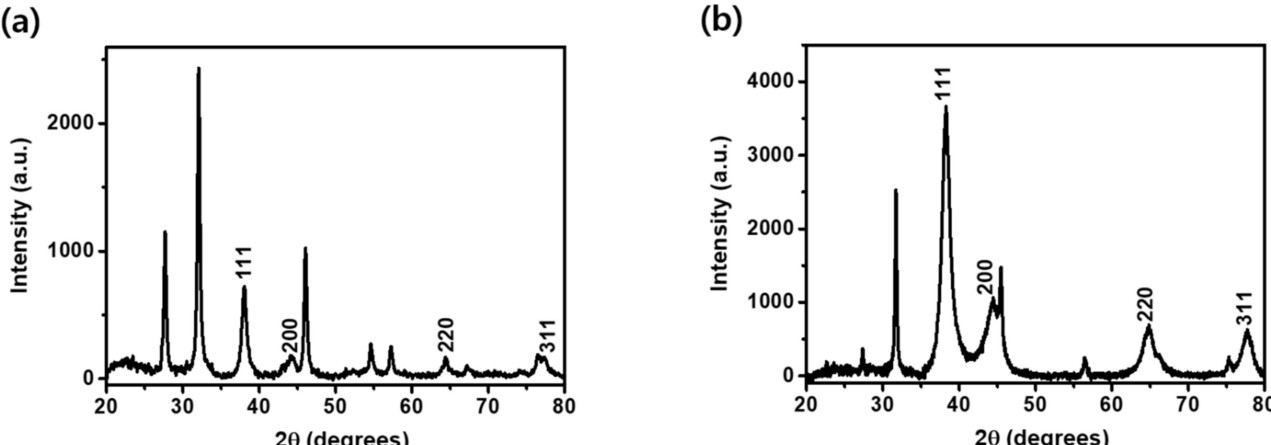

**Figure 5.** X-ray diffraction (XRD) reflection spectra for the (**a**) SS-AgNPs and (**b**) SS-AuNPs.

### 2.4. Fourier Transform Infrared (FT-IR) Spectral Analysis

FT-IR analysis was employed to reveal the SS extract's role, a capping agent. FT-IR spectra ensured chemical compounds of SS attached onto the bio-synthesized NPs. Figure **??** shows FT-IR spectra of the SS extract (a), AgNPs (b) and AuNPs (c). The FT-IR spectrum obtained from SS extracts indicated intensive bands. The bands observed around 3412 and 1397 cm$^{-1}$ corresponded to the −OH bond stretching in alcohols or polyphenol in SS. The intense peak at 1592 cm$^{-1}$ was attributed to N−H stretching in amine groups. Another peak obtained around 1079 cm$^{-1}$ was attributed to stretching of C−N in the aromatic ring. After green-synthesis, the NPs spectra exhibited various small shifts of peak position compared to SS. The peaks in SS-AgNPs at 3411, 1592, 1385 and 1078 cm$^{-1}$ and in SS-AuNPs at 3421, 1599, 1401 and 1079 cm$^{-1}$ derived from the above corresponding functional components. The peak around 1385 cm$^{-1}$ in SS-AgNPs was small and sharp only observed in spectra of the AgNPs corresponding to stretching vibration of N−O in nitro compounds. These observations suggested that the phytochemicals like polyphenol in SS extracts were attributed to capping metal NPs. The exact mechanism of nanoparticle synthesis requires further research although the phytochemicals have an antioxidant ability to donate electrons or hydrogen atoms [**?** ].

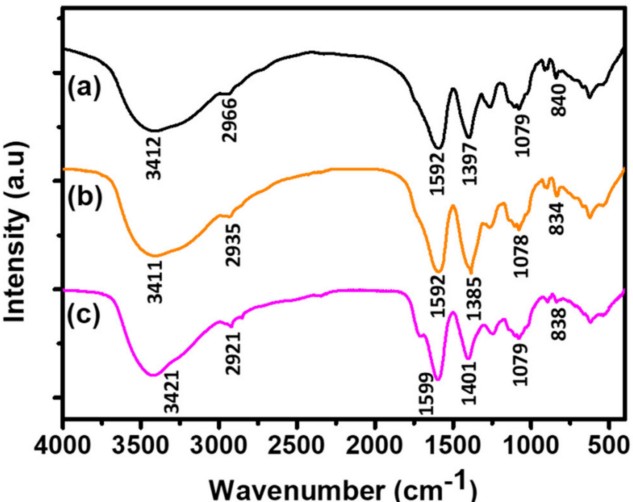

**Figure 6.** Fourier transform infrared (FT-IR) spectra of (**a**) SS extract, (**b**) SS-AgNPs, and (**c**) SS-AuNPs.

### 2.5. Catalytic Reduction of Sargassum serratifolium Silver Nanoparticles (SS-AgNPs) and Sargassum serratifolium Gold Nanoparticles (SS-AuNPs)

The catalytic activity of biogenic AgNPs and AuNPs on reduction of dyes were carried out with methylene blue (MB), rhodamine B (RB), and methyl orange (MO) under the condition of room temperature. Distilled water was added as the control instead of the nanoparticles. All decomposition of the dyes was plotted by the UV–Vis spectrophotometer every minute. The kinetics of reactions were analyzed spectrophotometrically. We assumed the catalytic reaction followed the pseudo-first order kinetics and calculated the rate constant, *k*, which is same value as the slope of each graph.

#### 2.5.1. Reduction of Methylene Blue

Methylene blue is a heterocyclic aromatic dye utilized in textile and medical treatment fields. It is a good antimalarial agent and remedy against methemoglobinemia [**?** ]. The maximum UV–vis spectrum peak of MB is present at 662 nm and reduced methylene blue called leucomethylene blue is colorless. The MB in natural water is toxic and the accumulation of MB in the human body can be harmful because of its carcinogenic and mutagenic to living beings [**?** ]. Figure **??**a represents the spectrum of a solution in which sodium borohydride and distilled water was added to methylene blue without a catalyst. After 40 min, the maximum peak was degraded slightly. On the other hand, the maximum peaks were almost degraded and the solutions became colorless both in 9 min as 150 µL of SS-AgNPs (Figure **??**b) or 10 µL of SS-AuNPs (Figure **??**c) was added instead of distilled water.

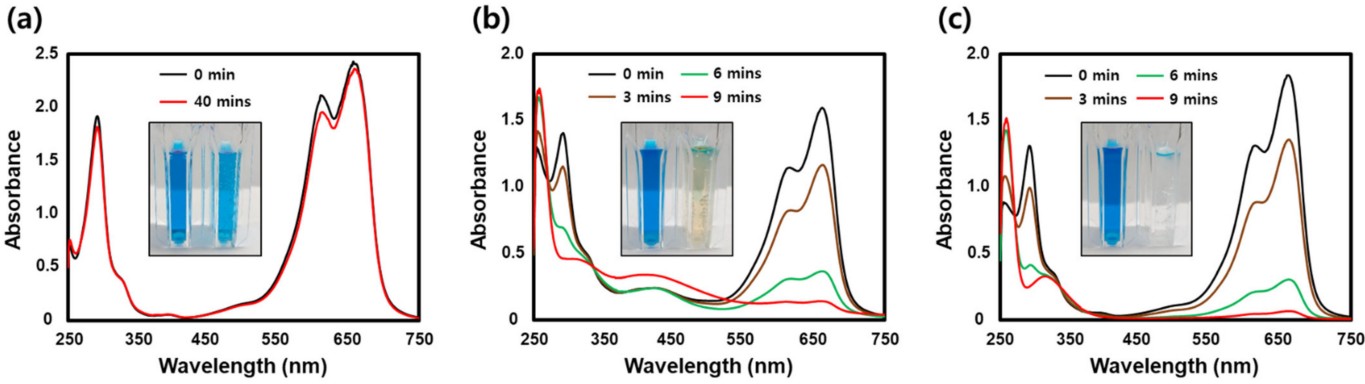

**Figure 7.** UV–Vis spectra of methylene blue (MB) dye reduction in the presence of different catalysts, (**a**) NaBH$_4$, (**b**) SS-AgNPs, and (**c**) SS-AuNPs.

MB was reduced and became colorless leuco form quickly due to the reduction potential and high surface-to-volume ratio of SS-NPs. As the electrons from BH$_4^-$ ions were transferred to MB by nanocatalyst, the reaction rate was accelerated reducing MB to leucomethylene blue [**?** **?** ]. The calculated reduction rate constants of MB were 0.3299 min$^{-1}$ (SS-AgNPs) and 0.4473 min$^{-1}$ (SS-AuNPs) exhibited in Figure **??**. Other detailed values are shown in Table **??**.

#### 2.5.2. Reduction of Rhodamine B

Rhodamine B is a water tracer fluorescent dye and has been broadly employed in textile and food industries [**?** ]. The maximum absorption peak of RB is at 553 nm. RB is degraded to amino compounds as the reduction products [**?** ]. It has been revealed that RB has carcinogenic and neurotoxic activity [**?** ]. The absorption measurement of RB with NaBH$_4$ and distilled water is shown in Figure **??**a. The surface plasmon resonance (SPR) peak at 553 nm was slightly lowered. However, with the addition of algae-mediated AgNPs (10 µL) and AuNPs (10 µL), the absorption measurement at 553 nm was completely decreased within 6 and 9 min and the solution became colorless, which is shown in Figure **??**b,c.

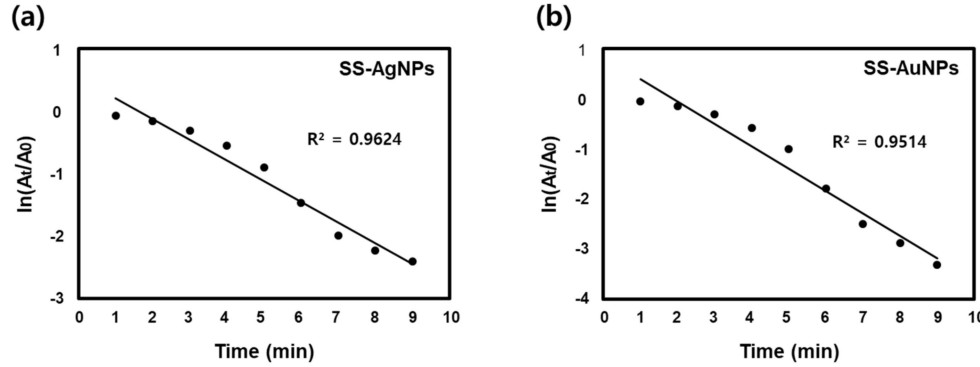

**Figure 8.** Plot of $\ln\frac{A_t}{A_0}$ versus time during MB reduction by catalysts, (**a**) SS-AgNPs and (**b**) SS-AuNPs.

**Table 1.** Values of methylene blue reduction.

| Dye | Catalyst | Reaction Time (min) | Rate Constant, k (min$^{-1}$) | Correlation Coefficient, $R^2$ |
|---|---|---|---|---|
| MB | SS-AgNPs | 9 | 0.3299 | 0.9624 |
| | SS-AuNPs | 9 | 0.4473 | 0.9514 |

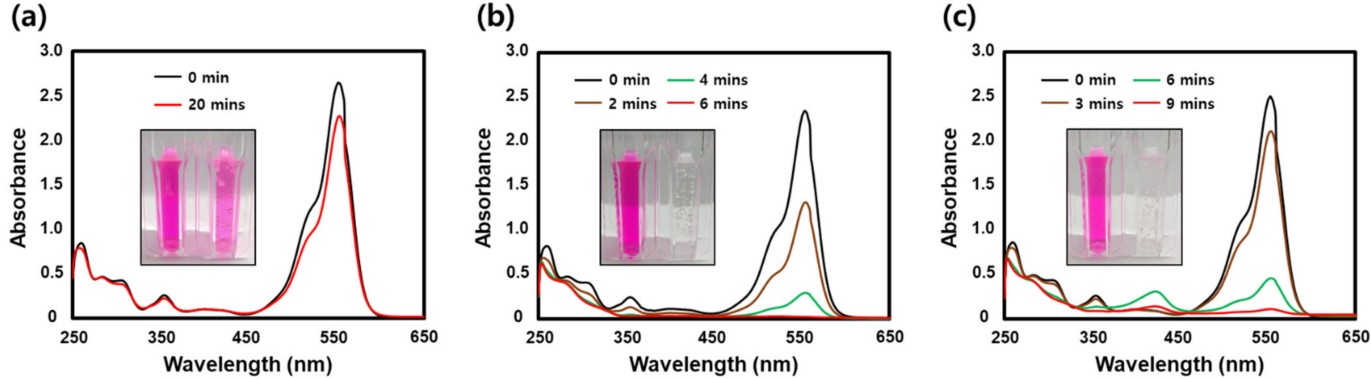

**Figure 9.** UV–Vis spectra of rhodamine B (RB) dye reduction in the presence of different catalysts, (**a**) NaBH$_4$, (**b**) SS-AgNPs, and (**c**) SS-AuNPs.

RB is reduced, decolorized and decomposed with the SS-NPs. The degradation rate constants of RB were 0.9131 min$^{-1}$ (SS-AgNPs) and 0.4607 min$^{-1}$ (SS-AuNPs) shown in Figure **??**. Other detailed values are exhibited in Table **??**.

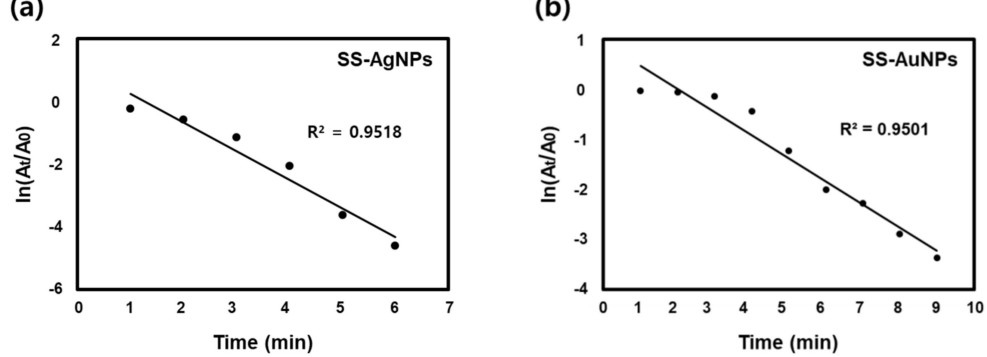

**Figure 10.** Plot of $\ln\frac{A_t}{A_0}$ versus time during RB reduction by catalysts, (**a**) SS-AgNPs and (**b**) SS-AuNPs.

**Table 2.** Values of rhodamine B (RB) reduction.

| Dye | Catalyst | Reaction Time (min) | Rate Constant, k (min$^{-1}$) | Correlation Coefficient, $R^2$ |
|---|---|---|---|---|
| RB | SS-AgNPs | 6 | 0.9131 | 0.9518 |
| | SS-AuNPs | 9 | 0.4607 | 0.9501 |

### 2.5.3. Reduction of Methyl Orange

Methyl orange (MO) has an azo group (R−N = N−R′) which play as chromophore [**?**]. The chromophore give MO a strong orange color. Therefore, MO can be used as a pH indicator and the maximum absorption band of it exists at 465 nm. MO was degraded into *N, N*-dimethyl-benzene−1, 4-diamine and 4-aminobenzenesulfonate. A large amount of MO that has not been decomposed can damage the environment and living being. Figure **??** shows the UV–Vis absorption spectrum of RB in the presence of NaBH$_4$ and distilled water. The spectrum was almost unchanged, meaning MO was hardly removed after 50 min. Meanwhile, in the presence of biosynthesized-AgNPs (20 μL) and AuNPs (20 μL) as a catalyst, the absorption band at 464 nm considerably degraded after 16 and 18 min (Figure **??**b,c) and MO turned colorless. It can be suggested that azo group of MO was decomposed due to SS-AgNPs and SS-AuNPs. The plots of ln $\frac{A_t}{A_0}$ versus time linear fits for catalytic reduction of MO with SS-AgNPs and SS-AuNPs are depicted in Figure **??**.

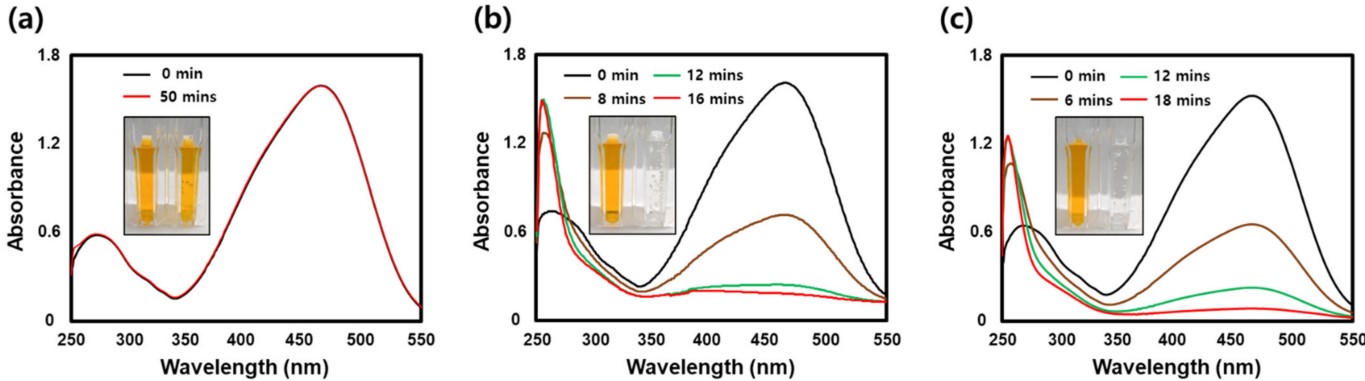

**Figure 11.** UV–Vis spectra of methyl orange (MO) dye reduction in the presence of different catalysts, (**a**) NaBH$_4$, (**b**) SS-AgNPs, and (**c**) SS-AuNPs.

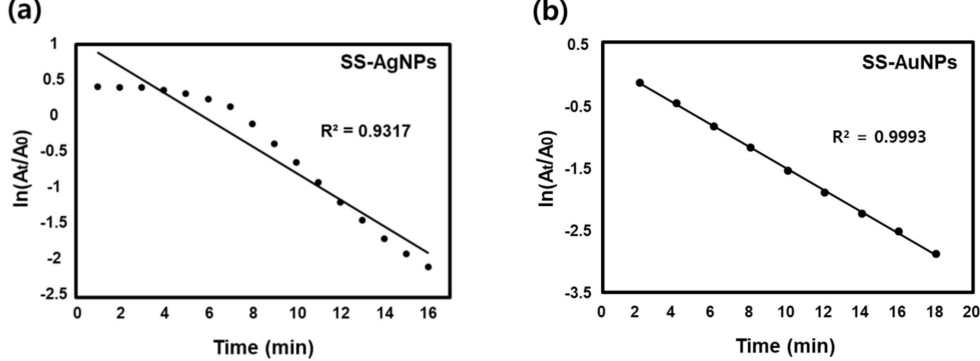

**Figure 12.** Plot of ln $\frac{A_t}{A_0}$ versus time during MO reduction by catalysts, (**a**) SS-AgNPs and (**b**) SS-AuNPs.

The calculated decomposition rate constants were 0.1580 min$^{-1}$ (SS-AgNPs) and 0.1712 min$^{-1}$ (SS-AuNPs). These were the smallest of the rate constants for the three dyes. The reason might be the strong nitrogen double bond (–N=N–) of azo group. Other details are provided in Table **??**.

**Table 3.** Values of methyl orange reduction.

| Dye | Catalyst | Reaction Time (min) | Rate Constant, k (min$^{-1}$) | Correlation Coefficient, R$^2$ |
|---|---|---|---|---|
| MO | SS-AgNPs | 16 | 0.1580 | 0.9317 |
| | SS-AuNPs | 18 | 0.1712 | 0.9993 |

*2.6. Mechanism of Dye Reduction*

All three dyes were reduced and turned colorless solutions by SS-AgNPs or SS-AuNPs used as the catalyst in short time of 18 min. During photocatalytic reduction, the electrons of the green-synthesized nanoparticles obtained energy from light and were excited from the valence band to conduction bands, and combined with oxygen and hydrogen from NaBH$_4$ to generate active radical species such as supper oxide anion ($O_2^{\bullet-}$) and OH radicals (OH●). The free radicals (supper oxide anion and OH radical) then broke the bonds of synthetic dyes.

The degradation of MB by gold-loaded hydroxyapatite nanoparticles under visible light was documented by Mondal et al. [?]. They showed that the generated active radical species broke the N−CH3, C−S, and C−N bonds of the dye molecules. According to Varadavenkatesan et al. [?], an electrons (e$^-$)-hole (h$^+$) pair between valence and conduction bands was created on the surface of ZnO nanoparticles. They reacted with water and oxygen to form highly reactive hydroxyl radicals (OH●) and superoxide radical anion ($O_2^{\bullet-}$) which were responsible for the fading of RB. Radini et al. [?] reported the possible mechanism of the photocatalytic degradation of MO. This was similar to the above documents. The strong radicals generated from FeNPs played an important role decomposing the –N = N– bonding and the complete mineralization of MO.

**3. Materials and Methods**

*3.1. Chemicals and Reagents*

*Sargassum serratifolium* extract was obtained from JEJU Technopark Inc. (Jeju, KR). Hydrogen tetrachloroaurate (III) trihydrate (HAuCl$_4$·3H$_2$O), silver nitrate (AgNO$_3$), methylene blue (MB), rhodamine B (RB), methyl orange (MO) and sodium borohydride (NaBH$_4$) were bought from Sigma-Aldrich Co. (St. Louis, MO, USA).

*3.2. Preparation of SS Extract*

*Sargassum serratifolium* extract was collected in Jeju Island, Jeju Province, Korea. Botanical identification was made by Mook Jae Lee (JEJU TECHNOPARK Inc., JEJU, KR), and a sample specimen was deposited at the herbarium of the Jeju Biodiversity Research Institute, Jeju, Korea. The SS was dried and was homogenized into a fine powder by an electric mixer (HMF-3100S, Hanil Electric, Seoul, Korea). The SS was prepared by dissolving the powder in 80% ethanol at room temperature, SS was then filtered and concentrated using a rotary vacuum evaporator, Buchi Rotavapor R-144 (Buchi Labortechnik, Flawil, Switzerland); 50 mL of SS extract was turned into powder using freeze-drying. This powder was held at −75 °C for sustainable use. For synthesis of NPs, the powder of algae became aqueous solution at a concentration of 4 mg/mL. The solution was filtered by a syringe filter (0.2 μm) for sterilization.

*3.3. Bio-Fabrication of SS-AgNPs and SS-AuNPs*

The temperature, reaction time and concentration of SS extract and metal precursor for biosynthesis of AgNPs and AuNPs was optimized before this work. First, 1 μL of aqueous AgNO$_3$ (1 M) solution was put into the filtered extract of 1 mL SS (2 mg/mL) and incubated in water for 15 min at 80 °C. After 15 min, the tube that contained the colloid was put in ice for 5 min. The color of suspensions went from yellow to dark orange. This suggested SS-AgNPs have synthesized. For synthesis of SS-AuNPs, 1 mL of the filtered SS extract (2 mg/mL) was added to 1 μL HAuCl$_4$·3H$_2$O (1 M) solution, following the

processes of AgNPs synthesis. The change to violet color corroborated that SS-AuNPs had been synthesized.

### 3.4. Characterization of SS-AgNPs and SS-AuNPs

UV–visible spectra of NPs green-synthesized using SS were recorded using an Ultrospec 6300 pro UV-Vis spectrophotometer (Amersham Biosciences, Buckinghamshire, UK) in 300–800 nm at wavelengths. The particle size of the dispersions and zeta potential were analyzed by a Zetasizer, Nano-ZS90 (Malvern Panalytical, Malvern, UK). A TALOS F200X (Thermo Scientific, Eugene, OR, USA) with EDX revealing HR-TEM images operating at 200 kV. The Ag and Au NPs were fixed onto a copper grid, Formvar/Carbon 200 Mesh (Electron Microscopy Sciences, Hatfield, PA, USA). X-ray diffractometer, X'Pert[3] Powder (XRD Empyrean series 2, PANalytical, Almelo, Nederland), was used to reveal the structural information of NPs. The instrument was operated at 40 kV with 30 mA in current and Cu K$\alpha$ radiation (1.540 Å) between 2θ° from 20° to 80° for analyzing the XRD pattern and crystal structure.

The FT-IR spectra in the range of 4000–400 cm$^{-1}$ were detected on Spectrum GX (Perkin Elmer Inc., Boston, MA, USA) using on operating through the potassium bromide pellet. The lyophilized nanoparticles were ground with KBr powder in a mortar pestle and analyzed to identify various functional groups in SS extract that have a role as a reduction agent for fabrication of SS-NPs.

### 3.5. Catalytic Study of SS-AgNPs and SS-AuNPs

The catalytic potential for reduction of dyes was studied using the UV-Visible spectrophotometer with ice cold NaBH$_4$ at the room temperature. NaBH$_4$ solution was freshly prepared prior to the experiments. Then, 2 mL aqueous solutions of MB (0.08 mM) and RB (0.05 mM) were added to 1 mL NaBH$_4$ (30 mM) and SS-AuNP (or SS-AgNP) solutions. Next, 2 mL of MO (0.1 mM) aqueous solution was mixed with 1 mL NaBH$_4$ (30 mM) and (AuNP or AgNP) solutions. SS-AgNP solution was added to the mixture at different volumes (150, 10, and 20 μL) of MB, RB, and MO, respectively. Meanwhile, the volume of the SS-AuNP solution added to the mixtures was 10 μL, except for the reduction of MO, where it was 20 μL. The amount of green-synthesized AgNPs and AuNPs and concentration of dyes and NaBH$_4$ for catalytic activities of SS-AgNPs and SS-AuNPs were optimized before these processes.

The kinetics of catalytic reaction were investigated by assuming the reaction process following the pseudo-first order law, with the formula:

$$-kt = \ln \frac{A_t}{A_0} \tag{2}$$

where $k$ is the rate constant, $A_t$ is the absorbance at time $t$, $A_0$ is the initial absorbance of dyes.

## 4. Conclusions

In this work, Ag and Au NPs using SS extract were synthesized at 80 °C via a facile and environmentally friendly synthetic method. The UV–Vis spectroscopy and EDX analysis substantiated the synthesis of SS-NPs. The size, zeta potential and morphology were revealed by DLS and HR-TEM. The XRD and SAED patterns confirmed that SS-AgNPs and SS-AuNPs were crystalline, FCC in structure. FT-IR spectra revealed that phytochemicals in SS extract have a role as stabilizing agents. These green-synthesized NPs have excellent performance of catalytic activities for reducing MB, RB and MO by NaBH$_4$ at room temperature. From the kinetic studies, the results suggested that the reduction of dyes followed a pseudo-first order and the rate constants ($k$) could be obtained. In conclusion, algae-based synthesized NPs have excellent catalytic activity and reactivity as cheap, desirable and environmentally safe nanocatalysts for the effective decomposition of

various dye pollutants. Therefore, they may be highly useful for water purification, various industries such as chemistry and textiles, and especially environmental remediation.

**Author Contributions:** Conceptualization, G.P. and S.Y.P.; Data curation, B.K. and W.C.S.; methodology, B.K. and W.C.S.; formal analysis, B.K. and W.C.S. and S.Y.P.; investigation, G.P. and S.Y.P.; Validation, B.K. and S.Y.P.; Visualization, B.K. and W.C.S.; writing—original draft preparation, B.K. and W.C.S.; writing—review and editing, G.P. and S.Y.P. All authors have read and agreed to the published version of the manuscript.

**Funding:** This research received no external funding.

**Acknowledgments:** This research was supported by the Basic Science Research Program through the National Research Foundation of Korea (NRF), which was funded by the Ministry of Education, Science, and Technology (NRF-2018R1D1A1B07047497 and NRF-2018R1D1A3B07047983).

**Conflicts of Interest:** The authors declare no conflict of interest.

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
