# Peer review of "Green Synthesis of Silver and Gold Nanoparticles via Sargassum serratifolium Extract for Catalytic Reduction of Organic Dyes"

_catalysts, doi:10.3390/catal11030347_

Round 1

Reviewer 1 Report

The authors reported the results for synthesis of Ag and Au NPs using bio-materials, such as Saragassum serratifolium. In addition, for test reactions, they performed the dye photo-reductions and showed the catalytic activities of the NPs. The results are well and will be suitable for the publication. However, the authors should revise the manuscript for its readability.

  1. The authors showed the FT-IR results for discussing the mechanism of the synthesis. For the mechanism, they used an expression that predicates. However, the variations in the FT-IR are small, and I think it is difficult to conclude the mechanism from only this data. So, the discussion of the detail mechanism should be a future work.
  2. The sizes of prepared NPs are large, and general sizes for the catalysts are under 10 nm although the prepared NPs shows the catalytic activities. Do the authors have any idea for decrease the NP sizes? If yes, please make a brief discussion on it.
  3. The authors estimated the NP sizes using the DLS. Please report the results estimated by TEM images. It is more directly for counting the NP size.
  4. The resolutions of Figures 2 and 3 are bad. Please improve them.
  5. The caption for Figure 2 is too cheap to understand what is shown in this figure. Please add the word “DLS.”
  6. The authors concluded the catalytic reactions are pseudo-first order. However, the ln-plots for some reactions have clear curves (not straight line), especially MO decomposition by Ag NPs. Please re-discuss these results.
  7. Did the authors perform Arrhenius plot? Can you report the activation energies?

Author Response

Thank you for reviewing and providing your comments on my manuscript.
I have revised my manuscript in accordance with the comments from the Reviewer, and my point-by-point responses are listed below.
1. The authors showed the FT-IR results for discussing the mechanism of the synthesis. For the mechanism, they used an expression that predicates. However, the variations in the FT-IR are small, and I think it is difficult to conclude the mechanism from only this data. So, the discussion of the detail mechanism should be a future work.
=> Thank you for your comment. I strongly agree with the statement that it is difficult to conclude the mechanism from only this FT-IR data. So I modified the line 20, 141, 153~156 and 313~314.
2. The sizes of prepared NPs are large, and general sizes for the catalysts are under 10 nm although the prepared NPs shows the catalytic activities. Do the authors have any idea for decrease the NP sizes? If yes, please make a brief discussion on it.
=> In accordance with the Reviewer’s comment, we conducted an experiment to synthesize nanoparticles by using 7 kinds of algae, including Sargassum serratifolium, as a reducing agent. The size of all nanoparticles synthesized with seven species of algae was around 10 nm. To reduce the size of the nanoparticles, we filtered all the extract solution with syringe filter. Also, the temperature, and concentration of SS extract and metal precursor for biosynthesis of AgNPs and AuNPs had been optimized before this study. 
3. The authors estimated the NP sizes using the DLS. Please report the results estimated by TEM images. It is more directly for counting the NP size.
=> Thank you for your comment. The average sizes of silver and gold nanoparticles shown in the TEM image are about 27.84 and 7.18 nm, respectively. We added this sentence in 111~112 line.    
4. The resolutions of Figures 2 and 3 are bad. Please improve them.
=> In accordance with the Reviewer’s comment, we improve them and change the Figure 2, 3 and 4.
5. The caption for Figure 2 is too cheap to understand what is shown in this figure. Please add the word “DLS.”
=> In accordance with the Reviewer’s comment, we added the word “DLS” at the caption for Figure 2.
6. The authors concluded the catalytic reactions are pseudo-first order. However, the ln-plots for some reactions have clear curves (not straight line), especially MO decomposition by Ag NPs. Please re-discuss these results.
=> Thank you for your comment.  A. In a catalytic reaction, there are several stages of reaction. Among them, a lot of energy is required to break the bond that represents the color of the dye, but the reaction until then proceeds slowly. When the bond is then broken, the reaction proceeds at an almost constant rate. So, in this reaction section, we think it would be a pseudo-first-order reaction. In MO decomposition, the initial reaction rate is estimated to be slower when using AgNPs as a catalyst than AuNPs. This is because methyl orange has a strong nitrogen double bond, and the size of AgNPs is larger than that of AuNPs, so that the specific surface area of AgNPs is smaller than that of AuNPs. In addition, the specific surface area of AgNPs can be smaller since AgNPs are relatively more aggregated than AuNPs as shown in the TEM figure 3 and 4. Last, MO is the highest concentration (0.1 mM) among three dyes. We also added the concentration value of MO, we had missed, in line 297. As a result, in our opinion, if the amount of catalyst increase, we expect that the initial reaction will be short and the overall reaction will proceed faster with the more constant reaction rate.
7. Did the authors perform Arrhenius plot? Can you report the activation energies?
=> Thank you for your comment. There was no experiment with variable temperature. We conducted a catalytic reduction experiment under the condition of fixing at room temperature with only UV-vis spectrophotometer and showed the reaction rate over time. Therefore, the activation energy cannot be obtained through the Arrhenius equation, but the activation energy can be obtained using Arrhenius plot and equation if the experiments with variable temperature are conducted. The paper did not mention the experiment was conducted at room temperature. We will add it in line 161 and 294. Thank you.

Reviewer 2 Report

The article entitled "Green Synthesis of Silver and Gold Nanoparticles via Sargassum serratifolium Extract for Catalytic Reduction of Organic Dyes" is interesting, bio-materials or bio-inspired materials are attracting more and more attention. However, I would like to make a few comments with the author.

First, in line 237 the symbol ")" is missing.

From my point of view, the introduction is not correct, since you are presenting catalytic results and the majority of the references used are not related to catalysis.

In the representation of the catalytic results, the authors assume a pseudo-order 1, but the discussion of the results is more complex. In all reactions, you see an induction period, where the reaction hardly takes place (the first few minutes). What is the reason for this induction period? Have the authors repeated the reactions? What is the reproducibility of the reactions? Have the authors performed catalyst reuse tests? If the authors have done reuse tests, do you observe the same induction phenomenon at the beginning of the reaction?

The authors should include more information about the reaction system, such as the reactor used, temperature, type of lamp, the intensity of light.

Author Response

The article entitled "Green Synthesis of Silver and Gold Nanoparticles via Sargassum serratifolium Extract for Catalytic Reduction of Organic Dyes" is interesting, bio-materials or bio-inspired materials are attracting more and more attention. However, I would like to make a few comments with the author.

First, in line 237 the symbol ")" is missing.

=> In accordance with the Reviewer’s comment, we have revised the in manuscript. (line 237).

From my point of view, the introduction is not correct, since you are presenting catalytic results and the majority of the references used are not related to catalysis.

=> In accordance with the Reviewer’s comment, we have revised the in manuscript.  

In the representation of the catalytic results, the authors assume a pseudo-order 1, but the discussion of the results is more complex. In all reactions, you see an induction period, where the reaction hardly takes place (the first few minutes). What is the reason for this induction period? Have the authors repeated the reactions? What is the reproducibility of the reactions? Have the authors performed catalyst reuse tests? If the authors have done reuse tests, do you observe the same induction phenomenon at the beginning of the reaction?

=> Thank you for your comment.  In a catalytic reaction, there are several stages of reaction. Among them, a lot of energy is required to break the chemical bond that represents the color of the dye, but the reaction until then proceeds slowly. When the bond is then broken, the reaction proceeds at an almost constant rate. So, in this reaction section, we think it would be a pseudo-first-order reaction. We repeated the same experiment 3 times and the same phenomenon was observed at the beginning of the reaction. However, the reuse test was not performed because the colloidal catalyst used in the experiment was added to the dye and NaBH4 aqueous solution in liquid form. In our opinion, if the amount of catalyst increase, we expect that the initial reaction will be short and the overall reaction will proceed faster with the more constant reaction rate.

The authors should include more information about the reaction system, such as the reactor used, temperature, type of lamp, the intensity of light.

=> In accordance with the Reviewer’s comment, we used Ultrospec 6300 pro UV/Visible Spectrophotometer for the catalytic activity. The space in which the experiment was conducted was inside the device and no external light came in. Also, all the experiments were conducted at the room temperature. The lamp were deuterium and tungsten lamps. The specifications of the device is as follows. http://www.promix.ru/manuf/ge/sf/sf.pdf

Round 2

Reviewer 1 Report

The manuscript has been revised carefully by the authors. The paper is well presented. I recommend the acceptance of this paper.